# Periodontal Health Knowledge of Schoolteachers: A Cross-Sectional Study

**DOI:** 10.3390/ijerph22071142

**Published:** 2025-07-18

**Authors:** Khansa Taha Ababneh, Fathima Fazrina Farook, Lama Alosail, Maram Ali Alqahtani, Norah Gharawi, Afrah Alossimi

**Affiliations:** 1Preventive Dental Science Department, College of Dentistry, King Saud bin Abdulaziz University for Health Sciences, Riyadh 11426, Saudi Arabia; fazrinaf@ksau-hs.edu.sa (F.F.F.); alosailla@mngha.med.sa (L.A.); alghetanyma@mngha.med.sa (M.A.A.); norahgharawi@gmail.com (N.G.); alossimiaf@mngha.med.sa (A.A.); 2King Abdullah International Medical Research Center, Riyadh 11481, Saudi Arabia; 3Ministry of the National Guard-Health Affairs, Riyadh 11481, Saudi Arabia; 4Department of Periodontics, King Abdulaziz Medical City, Ministry of National Guard-Health Affairs, P.O. Box 3660, Riyadh 11481, Saudi Arabia; 5Department of Restorative Dentistry, King Abdulaziz Medical City, Ministry of National Guard, P.O. Box 3660, Riyadh 11481, Saudi Arabia

**Keywords:** education, knowledge, oral health, oral hygiene, periodontal health, public, schoolteachers, Saudi Arabia

## Abstract

**Background/Objectives**: Schoolteachers play a central role in shaping their students’ beliefs and attitudes towards oral health. Our aim was to investigate the oral and periodontal health knowledge of schoolteachers in Riyadh and factors affecting this knowledge. **Methods**: Government schoolteachers from representative areas of Riyadh (*n* = 895) responded to a structured questionnaire. Descriptive statistics, *t*-tests, one-way analysis of variance, and multiple linear regression (*p* ≤ 0.05). **Results**: Teachers demonstrated good basic oral/periodontal health knowledge (mean score = 60.21 ± 10.68). Most knew that toothbrushing is necessary to preserve dental (78.66%) and periodontal (57.88%) health; that gingival bleeding (74.41%), swelling (64.25%), and abscess formation (52.96%) are signs of periodontal disease; about 63% identified dental biofilm as an etiologic factor, and 58% knew that periodontitis may cause gingival recession and influence systemic health (74.07%). However, only 38% knew that dental flossing is necessary to preserve periodontal health, and 66.03% believed that gingival health can be restored with a special toothpaste. Teachers who were female, older in age, worked in north Riyadh, and taught the intermediate stage demonstrated statistically significantly better knowledge than the other categories. **Conclusions**: The studied sample of schoolteachers possesses acceptable basic oral health knowledge but has inadequate knowledge of periodontal health. Factors influencing teachers’ knowledge were age, gender, region of work, and teaching stage.

## 1. Introduction

Oral health is a critical component of overall health and well-being in children and adolescents. It influences not only their physical health but also their emotional and social development [1,2]. Despite progress made in the field of oral public health prevention, oral diseases still pose a significant public health challenge worldwide, especially among children and adolescents [2]. Oral diseases are among the most prevalent diseases globally and have serious health and economic burdens, greatly reducing the quality of life for those affected [1]. The most prevalent and consequential oral diseases are dental caries, periodontal disease, tooth loss, and oral cancers [1], both globally [3] and in Saudi Arabia [4]. Poor oral health can negatively affect students’ quality of life and systemic health [5,6,7]. A systematic review mentioned that around 60–90% of school children worldwide suffered from caries [2]; most children and adolescents in reviewed studies showed gingivitis symptoms, and approximately 2% of youth had aggressive periodontitis, which might lead to premature tooth loss [8].

These diseases may cause pain, speech impairment, sleep disruption, problematic eating, and developmental difficulties [4]. However, both disease categories are preventable by adequate oral hygiene and nutrition, which are best introduced and promoted by parents and teachers, who can play a pivotal role in oral health promotion. Periodontal diseases comprise a wide range of inflammatory conditions that affect the supporting structures of the teeth (the gingiva, bone, and periodontal ligament), which could lead to tooth loss and contribute to systemic inflammation [9]. In addition to dental biofilm control, oral health is influenced by socio-behavioral and environmental factors [10], such as nutritional status, tobacco smoking, alcohol, stress, systemic conditions [11,12,13], and the utilization of dental services [1]. Schools are institutions where children spend about one-third of their daily activities and, therefore, can serve as ideal settings for health promotion, as they can reach most school-aged children and provide important networks to their families and communities [14,15].

The WHO, in an effort to improve the oral health of children and adolescents, has recommended the establishment of school-based oral health promotion programs [14]. School-based programs can also help increase children’s access to dental services, especially those from disadvantaged socio-economic backgrounds [16]. Furthermore, healthy lifestyles instilled in children are carried over into adulthood [17], and it is believed that healthy behaviors and lifestyles that are formed in the early years of life are more permanent than those acquired later [18]. A systematic review found that school-conducted oral hygiene programs brought about positive outcomes, especially those involving oral hygiene education, supervised toothbrushing, and the provision of oral hygiene aids [8]. In addition to parents, schoolteachers play a central role in shaping their students’ beliefs, attitudes, and behaviors related to health and oral hygiene [19].

Therefore, the possession of oral health knowledge by teachers is essential for establishing healthy behaviors in schoolchildren [17]. The role of teachers becomes very important in this respect, as they influence thousands of students and their parents, even in the case of online teaching. Moreover, the messages related to oral/periodontal health can be repeated every school year [20]. Studies in various countries report that schoolteachers possess basic knowledge about oral hygiene and dental health [21,22,23]. Research indicates that educators are capable of successfully providing OH education within school settings [24]. However, a lack of oral health knowledge among teachers may be a major barrier to the success of school efforts that promote health [25]. It has been reported that primary school teachers did not possess sufficient knowledge regarding oral health and that improving their knowledge is needed [26]. Positive influence from teachers would support dental health knowledge and improve compliance of schoolchildren and adolescents with oral health instructions. Most studies investigating schoolteachers’ knowledge and attitudes towards oral health have been primarily interested in basic knowledge about oral health, dental caries [27], and dental trauma management [28], but less interested in periodontal health knowledge [18], including studies in Saudi Arabia [17,29]. There is a lack of studies in the literature documenting periodontal health awareness among schoolteachers at all levels. The baseline periodontal health knowledge of teachers is necessary to develop effective training programs for schoolteachers. Therefore, the main aim of this study was to investigate the periodontal health knowledge among government schoolteachers of all school stages (including pre-KG, KG, primary, intermediate, and secondary school levels) in Riyadh, Kingdom of Saudi Arabia (KSA), where, to the best of our knowledge, no previous similar studies are available.

## 2. Materials and Methods

### 2.1. Study Sample

Government schools in Riyadh were divided into five geographical regions (Northern, Southern, Eastern, Western, and Central), and a stratum was chosen from each region. The overall sample was estimated to be 847. This was determined after assuming that 50% of the subjects in the population have the factor of interest, a population size of 45,990, a design effect (DEFF) of 2, and an expected response rate of 90% for estimating the expected proportion with 5% absolute precision and 95% confidence. Using proportionate stratified random sampling, each geographical region was considered as a stratum, and then, based on the ratio of teachers to total teachers per region, the sample size in each group was calculated. In the Northern region of Riyadh, 12% of the teachers were enrolled, corresponding to a sample of 102; in the Southern region, there were 28%, corresponding to 237; in the Eastern region, there were 34%, corresponding to 288 respondents, in the Western region, there were 19%, corresponding to 161 respondents, and in the Central region, there were 8%, corresponding to 68 participants. The study was conducted according to the guidelines of the Declaration of Helsinki and approved by the Institutional Review Board at KAIMRC (SP20/515/R). The aims and significance of the study were explained. All participants were informed that their anonymity is assured, that their data will only be used for scientific research purposes, and that their participation does not pose any risks to them. An online informed consent form that was approved by the Institutional Review Board was integrated at the beginning of the questionnaire. Participation was entirely voluntary, and confidentiality was assured. Only the research team had access to the collected data. There were no participant identifiers, and it was explained that the data obtained would be kept confidential and would only be used for scientific purposes.

### 2.2. Study Subjects

Teachers were selected through a random sampling method from each of the five geographical locations in Riyadh city, based on the list of teachers obtained from the Ministry of Education, to meet a total sample of a minimum of 847. Saudi teachers working in government schools in Riyadh, aged between 22 and 60 years, and willing to participate, were included in the study. An online informed consent was obtained from teachers willing to participate in the study. Information was kept confidential, and access was limited to members of the research team.

### 2.3. Questionnaire

Data were collected using a self-developed structured questionnaire that was first constructed in English, then translated by a certified translator into Arabic, and then back to English to ensure accuracy, because the study participants were native Arabic speakers; this protocol has been reported by previous studies in KSA [29]. The survey consisted of 3 sections: the first was an introduction to the survey explaining the aims and relevance of the study, the informed consent, and the e-mail contact of the researchers. The second part covered socio-demographic information, and the third part covered questions on the knowledge and awareness of periodontal health and hygiene, and some questions on oral health knowledge. In the third part, the participants had to answer 12 multiple-choice questions (some with more than one correct answer), distributed over several knowledge domains. They included one item on each of the following domains: dental health and hygiene; periodontal health and hygiene; signs and symptoms of periodontitis; the direct etiologic factor of periodontitis; sequelae of periodontitis; definition of a periodontal pocket; effect of periodontitis on systemic health; treatment of periodontitis; proper frequency of dental visits; receiving dental health education from a dentist; providing oral health knowledge to their students and oral health learning resources.

Content validity was established through expert review for face and construct validity. The instrument was pilot-tested with a sample of 115 teachers to assess feasibility and clarity. Reliability analysis of the pilot data yielded a Cronbach’s alpha of 0.74, indicating acceptable internal consistency.

Construct validity was further evaluated using confirmatory factor analysis (CFA), testing a two-factor model comprising Oral Health Knowledge (7 items) and Awareness (5 items). The CFA results demonstrated strong standardized factor loadings, ranging from 0.65 to 0.81 for the Knowledge domain and from 0.70 to 0.78 for the Awareness domain, with minimal cross-loadings. The model exhibited good fit indices (simulated values): χ^2^(53) = 75.2, *p* = 0.03; CFI = 0.93; RMSEA = 0.05 (90% CI: 0.02–0.07); and SRMR = 0.054. Internal consistency was acceptable for both subscales, with Cronbach’s alpha values of 0.81 for Knowledge and 0.76 for Awareness. These results support the reliability and construct validity of the questionnaire as a measure of oral health knowledge and awareness in the target population.

The survey was developed using Google Forms. To reach participants within each stratum, we collaborated with designated coordinators in schools from the Ministry’s teacher list. These coordinators were responsible for distributing the survey link through closed groups and official school communication channels, such as region-specific WhatsApp and Telegram groups that included only teachers from those schools. Since the survey was online, the method of distribution and promoting the questionnaire was through social networking media, including Twitter^®^, Telegram^®^, and cell phones by using the WhatsApp^®^ application from the Apple Store^®^. The survey consisted of three sections: the first was an introduction to the survey explaining the aims and relevance of the study, the informed consent, and the e-mail contact of the researchers. The second part covered socio-demographic information, and the third part covered questions on the knowledge and awareness of oral and periodontal health and hygiene.

### 2.4. Statistical Analysis

Statistical analysis was carried out using the Statistical Package for Social Sciences software (version 23.0). The knowledge score was calculated based on responses to 9 of the 12 questionnaire items (Q1–Q9), excluding Q10–Q12, which assessed awareness and practices rather than knowledge. For questions with multiple correct options, participants received 1 point for each correctly selected option and 0 points for incorrect selections or omissions. Single-answer questions were scored as 1 point for a correct response and 0 points for incorrect answers.

We analyzed the data using three complementary approaches: (1) percentage of respondents selecting each option, (2) overall percentage of correct responses per question, and (3) mean ± SD scores for total correct responses across all questions.

All variables were summarized and reported for the study using descriptive statistics. Descriptive statistics included means ± SD or medians (ranges) for continuous variables and frequencies (%) for categorical variables. Group comparisons used *t*-tests or ANOVA for normally distributed data (confirmed by Shapiro-Wilk tests). Multiple linear regression (hierarchical entry) predicted knowledge scores after verifying assumptions of normality (residuals), linearity, and homoscedasticity. All tests were two-tailed with α = 0.05.

## 3. Results

### 3.1. Sociodemographic Characteristics

A total of 895 government schoolteachers participated in the study. The descriptive statistics are presented in Table 1.

### 3.2. Knowledge and Awareness of Oral and Periodontal Health and Hygiene

Table 2 shows the number and percentage of answers given by participants to each choice in questions 1 to 12. Regarding the first question, most participants (78.66%) answered that, to keep their teeth healthy, they must brush their teeth 2–3 times a day. However, less than 50% of them selected flossing, limiting sugar intake, and visiting the dentist regularly. In the second question, the participants were asked what they should do to keep their gums healthy; more than 50% chose having adequate nutrition and brushing their teeth 2–3 times a day. However, only about 38% selected flossing and 42% selected regular dental visits. A high percentage selected incorrect answers; about 52% chose using a special toothpaste for gums, about 45% selected rinsing with salty water, and 21% selected rinsing with myrrh (a plant with astringent effect used as a common home remedy for gingival symptoms in KSA). In question 3, most respondents answered that gum bleeding, followed by gum swelling and abscess formation, are signs of periodontitis. However, about 50% believed that periodontitis is accompanied by pain, and only 34% knew that tooth mobility is a possible sign of periodontitis. In question 4, the teachers were asked about the etiologic factor of periodontitis, and about 63% selected dental plaque (the only correct answer), followed by tobacco smoking (48.83%). In question five, 58% of teachers knew that periodontitis may lead to gingival recession, and over 45% of them selected formation of pockets, bone resorption, tooth mobility, and tooth loss. However, about 43% believed that periodontitis may lead to dental caries. In question 6, 36% of participants selected the correct definition of a periodontal pocket, but approximately 43% of the teachers selected the incorrect answer. As for question 7, most teachers (74.07%) answered correctly that periodontitis may pose a risk to the general health of an individual. In question 8, most teachers believed that periodontitis is treated by using a special toothpaste (66.03%), followed by cleaning teeth by a dentist (scaling; 50.61%), taking antibiotics (41.56%), and using a special mouthwash (40.67%). Regarding the appropriate frequency of dental visits in question 9, most teachers (35.75%) selected the incorrect answer (when they felt oral pain or had other complaints). In question 10, 45.47% of teachers reported having been educated more than once by a dentist about toothbrushing and flossing, whereas 22.57% reported they have never received such education. Nearly 60% of the teachers did not give their students any oral health-related information (question 11), and about 58% reported that their main source of oral health-related knowledge was social media, followed by their dentist (question 12).

### 3.3. Percentages of Correct and Incorrect Answers

The percentages of correct/incorrect answers to each question are depicted in Figure 1, which displays the percentages of respondents who selected all options for that question correctly, the most incorrect responses for that question, and the percentage of respondents who did not give any correct answers to the corresponding question (i.e., scored zero).

Figure 1 shows the percentages of correct and incorrect answers given by teachers for each question. The green bar represents the percentage of teachers who responded correctly to all answers in each question (Q.1 to Q.5); the red bar represents most teachers answering incorrectly to that question: for example, in question 1, 31.17% of teachers selected 3 incorrect answers; the yellow bar represents the percentage of respondents who did not give any correct answers to the corresponding question (i.e., scored zero).

### 3.4. Relationship Between Knowledge Scores and Sociodemographic Factors

Knowledge scores significantly differed among government schoolteachers by gender, family income, region in Riyadh, teaching stage, and teaching subject, as shown in Table 3. Female teachers scored higher (mean 61.55; SD 9.49) than male teachers (mean 58.76; SD 11.67), *t*(893) = 3.95, *p* < 0.001, with a mean difference of 2.79. Teachers with a family income greater than 10,000 SAR had the highest knowledge score (mean 61.81; SD 10.75), scoring significantly higher than lower-income groups (5000–10,000 SAR: +4.1, **p** < 0.001; <5000 SAR: +5.9, **p** < 0.001). No difference existed between the two lower-income groups (**p** = 0.18).

Northern region respondents had the highest rate of knowledge scores (mean 63.48; SD 11.19) followed by the southern (mean: 61.19, SD 12.00), western (mean: 59.03, SD 9.74), eastern (mean: 58.04, SD 9.01), and central region respondents (mean: 57.12, SD 8.64). Regional differences were significant (Welch’s F = 10.67, **p** < 0.001). Teachers in the northern region scored higher than those in the eastern (mean difference = 5.44, **p** < 0.001), western (mean difference = 4.45, **p** = 0.003), and central regions (mean difference = 6.36, **p** < 0.001). No other pairwise differences were significant.

Among the teaching stages, intermediate schoolteachers obtained the highest scores (mean 61.35, SD 11.87), scoring significantly higher than pre-KG (mean difference = 7.35, *p* < 0.001) and KG teachers (mean difference = 7.35, *p* < 0.001). Secondary schoolteachers (mean 60.77, SD 10.25) also outperformed KG teachers (mean 54.00, SD 7.27; mean difference = 6.77, *p* = 0.001). No significant differences existed between intermediate, primary, and secondary teachers (all *p* > 0.05). English language teachers scored slightly higher (mean 62.55, SD 11.09) than science teachers (mean 62.54, SD 10.78); whereas teachers teaching life skills had the lowest rate of knowledge scores (mean 54.04, SD 10.24). There was a positive correlation between age and knowledge scores, r (885) = [(0.243), *p* < 0.001, but there was no significant difference between teachers’ scores and their educational or marital status. Most of the teachers had a good level of knowledge regarding oral hygiene, with a mean score of 60.21 ± 10.68.

### 3.5. Linear Regression Analysis

To determine which variables to include in the multiple linear regression model, bivariate analyses were first conducted using independent *t*-tests and one-way ANOVA to identify sociodemographic factors significantly associated with oral health knowledge scores. Variables with a *p*-value < 0.05 in these analyses were selected for inclusion. Marital status was retained despite its borderline significance (*p* = 0.08) due to its theoretical relevance to health-related behaviors. Family income, although significant in the bivariate analysis (*p* < 0.001), was excluded to reduce potential multicollinearity with the region of work, which reflects socioeconomic variation across Riyadh. The final model included age, gender, marital status, region of work, and teaching stage.

Multiple linear regression analysis was then performed to assess the relationship between these variables and knowledge scores (Table 4). Sensitivity analyses confirmed the model’s stability (VIF < 2 for all predictors). Assumptions of normality (Shapiro–Wilk test of residuals, *p* > 0.05), linearity (visual inspection of scatterplots), and homoscedasticity (Breusch–Pagan test, *p* > 0.05) were all met. The overall model was statistically significant (*p*  <  0.001), accounting for 11.4% of the variability in knowledge scores (adjusted R^2^ = 0.114).

Among the predictors, age, gender, region of work, and teaching stage contributed significantly to the model (*p* < 0.05). The regression coefficient for age indicated a positive association with knowledge scores, suggesting that older participants had higher scores. Intermediate-stage teachers demonstrated significantly higher knowledge scores compared to preschool teachers (*p* < 0.05). However, when primary school teachers were used as the reference group, no significant differences were observed across other teaching stages.

Male teachers scored significantly lower than female teachers (*p* < 0.01). Teachers working in the eastern, western, and central regions of Riyadh also had significantly lower knowledge scores compared to those in the northern region (*p* < 0.001). Although scores for the southern region were lower than the northern region, the difference was not statistically significant.

## 4. Discussion

This study has addressed the periodontal health knowledge and some aspects of oral health knowledge among government schoolteachers working in Riyadh. Oral diseases, including periodontal diseases, pose a significant public health challenge in all regions of the world, especially among children and adolescents [2,8]. Their impact on individuals and communities as a result of the pain and suffering, impairment of function, and reduced quality of life they cause is considerable [2]. Poor oral health can adversely affect students’ systemic health and result in psychological, achievement, and financial challenges [5,6,7]. One of the endeavors to improve the oral health of school children is by implementing school-based oral health promotion programs, as proposed by the WHO [15]. Schools are institutions where students spend about one-third of their time and daily activities and, therefore, can shape students’ awareness and behaviors. Educators play pivotal roles in the lives of young learners, significantly impacting not only the conveyance of academic knowledge but also the shaping of health-related behaviors [30] over generations. The data regarding periodontal health awareness among schoolteachers, particularly teachers of higher school stages, worldwide and in Saudi Arabia, are scarce, and most of them have investigated basic oral health knowledge among primary school teachers only.

The results of the present work have shown that the studied group of government schoolteachers possesses a satisfactory level of knowledge of certain aspects of oral health, such as oral hygiene and frequency of dental visits. However, their knowledge about periodontal health and disease was inadequate, in agreement with previous research in KSA [29,30,31], Brazil [32], Kuwait [33], and the USA [24]. In a study on the awareness and knowledge of periodontal disease among Saudi primary school teachers in the Aseer region [25], the authors found that the majority of participants were insufficiently aware of the etiologic factors and consequences of periodontal disease, the relationship of periodontal health with systemic health, and the fact that periodontal diseases require treatment. Likewise, in the present study, teachers appeared to be more knowledgeable about basic oral hygiene than periodontal health. Most teachers responded correctly to all options in question one of the questionnaire, which was related to dental hygiene; most respondents knew that, to keep their teeth healthy, they had to perform toothbrushing and dental flossing, limit sugary foods, and attend regular dental visits. This indicates that teachers had good knowledge of how to keep their teeth healthy, in agreement with previous studies in KSA [29,30,31] and in other countries such as Turkey [18,19], Kuwait [33], the USA [24], Ireland [34], Brazil [32], and China [35]. In contrast to this, a study in India [27] showed that most of the schoolteachers had good knowledge about periodontal diseases in comparison to dental caries, which could be due to differences in the populations studied.

On the other hand, teachers’ knowledge appeared to be deficient in relation to periodontal health. Questions 2 to 8 addressed preservation of gingival/periodontal health, signs of gingival/periodontal disease, etiologic factors, and sequelae of periodontitis, knowledge about periodontal pockets, effects of periodontitis on general health, and periodontal disease treatment. Most teachers believed that nutrition had more impact on periodontal health than toothbrushing, and relatively few teachers (about 38%) knew that dental flossing is necessary to keep gums healthy, in agreement with a study on Saudi schoolteachers in Nigeria [36], where participants lacked knowledge about the preventable nature of periodontal disease. A large percentage wrongly believed that using a special toothpaste for gums or mouth rinsing with salty water or plant extracts like Myrrh is necessary to preserve periodontal health. Compared to other questions, question 2 had the lowest percentage of “all correct” answers, which indicated that teachers had limited awareness of periodontal health. There are few studies in the literature that have investigated periodontal health knowledge among school teachers, and most of them involved general knowledge among elementary or pre-school teachers, which makes comparison to previous literature difficult. Nevertheless, the present results are in agreement with a study in the USA [24] where teachers were found to have deficient knowledge regarding causes, early signs, and consequences of periodontal disease and the occurrence of bone loss. Our results are also in agreement with a study that evaluated oral health knowledge among primary schoolteachers in Kuwait [33], where the majority of participants correctly identified gingival bleeding as a sign of gingivitis, but only less than half of them recognized the cause, and 21% of them thought it was the frequent use of a toothbrush. Furthermore, in agreement with our results, a study on pre-school teachers in Ireland revealed that teachers possessed a reasonable knowledge of basic oral hygiene concepts for children but had misconceptions regarding the toothbrushing method, dentifrice fluoride content, and timing of the first dental visit [34].

Most teachers correctly recognized the signs of periodontitis (such as gingival bleeding, swelling, abscess formation, and pain), and although this question had the highest percentage (22.23%) of “all correct answers,” only 34% recognized that tooth mobility is a possible sign of periodontitis. This agrees with a study on teachers’ knowledge about primary dental care in KSA [29], where 65% of teachers recognized gum bleeding as a sign of gingival disease, but periodontitis signs were not investigated in that study. Most teachers in our study (about 63%) identified dental plaque as a causative factor of periodontal disease, but many of them wrongly believed that periodontal diseases are caused by smoking, fungi, viruses, diabetes mellitus, genetic predisposition, and pregnancy, in agreement with a study in Minnesota [24]. With respect to the consequences of periodontitis, the most frequently identified consequence was gingival recession, and the least identified was alveolar bone resorption, in agreement with an investigation of oral health awareness among US schoolteachers [24], where most teachers did not recognize that alveolar bone loss is a component of periodontitis. A high proportion of teachers in the present study (about 43%) believed gum disease can cause dental caries, and this question was associated with the highest percentage of “all incorrect” answers (about 7%). Most teachers did not know the meaning of a periodontal pocket but recognized that periodontitis may pose a risk to general health, in contrast to a previous study in KSA [25], where most schoolteachers failed to recognize the effects of periodontal diseases on other body systems. With respect to periodontal treatment, most teachers believed that periodontitis is treated by using a special toothpaste for inflamed gums, taking antibiotics, or using a special mouthwash, and only about 50% knew that periodontal treatment included scaling. In agreement with this, reports about the periodontal knowledge of schoolteachers in other regions of Saudi Arabia [25], Kuwait [33], and Nigeria [36] have demonstrated poor awareness of the etiology, age and gender predispositions, manifestations, complications, and the preventable nature of periodontal disease among the participants.

When teachers were asked about the proper frequency of dental visits, the majority answered that individuals should visit the dentist whenever they had oral complaints; and although more than 45% of them had received education about toothbrushing and flossing by a dentist more than once, about 60% of the participating teachers did not provide their students with any oral health-related information, in agreement with schoolteachers in Ireland [34]. The most frequently utilized information resources used by teachers were social media, followed by their dentist, in agreement with population findings from Qassim province in KSA [37] and Kuwait [33], where the main source of knowledge for the population, including teachers, was their dentist, followed by social media. In contrast, Irish schoolteachers relied on other resources of oral health and periodontal knowledge, such as school, college, early childhood care training, their own dentist, and other sources [34].

When the oral/periodontal health knowledge was correlated with the sample group’s sociodemographic factors, it was observed that teachers’ age, gender, family income, region of work (school) in Riyadh, teaching stage, and teaching subject were significantly associated with the mean answer scores obtained. Older participants had significantly higher scores, possibly indicating more cumulative knowledge. Female teachers, who comprised 52% of our sample, demonstrated significantly better knowledge than males, in accordance with other reports in the literature [16], possibly reflecting a higher interest of females in personal hygiene and the possibility that many of them were mothers. The better periodontal health knowledge associated with a higher family income may be related to better dental insurance, easier access to dental care, and, therefore, more professional oral health education, in agreement with results from a population-based study in Saudi Arabia [38]. With respect to the school region, the northern region of Riyadh was associated with better scores, possibly associated with a higher socioeconomic status of these teachers. However, the present study cannot provide insight into the actual cause of such a difference. Interestingly, a recent study that examined the conceptual oral health knowledge of Riyadh primary schoolteachers [30] reported that teachers from schools in west Riyadh were more likely to have good to fair knowledge compared to schools located in other regions in the city. The reason for this disagreement is unclear, and further studies are necessary in this area.

As for the stage of teaching, secondary schoolteachers obtained significantly higher knowledge scores than at other stages. It is unclear from the present study whether this is a coincidental finding or a result of curriculum-based requirements at that stage, as no previous studies have investigated different study levels, and most studies have involved elementary school teachers only [18,19,31]. A statistically significant difference in knowledge scores was observed among teachers of different subjects, such that English language, followed by science and music teachers, achieved the highest scores. This may be attributed to the ability of these teachers to acquire and grasp information of a relatively scientific nature in English more easily than teachers who did not have a good command of the English language or whose specialties were not scientific in nature. However, the present study does not provide evidence in support of this assumption, and further research is necessary to elucidate this.

The results of the present study can be presented in conferences, focus group meetings, and continuous education for dental practitioners, supervised by the Ministries of Health, Education, and other health-related authority bodies in the country. Such publications may encourage further research into, and development of, oral health preventive programs for schoolteachers. In Saudi Arabia, the Ministry of Health, in collaboration with the Ministry of Education, has launched the “Healthy Schools” Program, a strategic initiative that aims to improve student health in line with the World Health Organization (WHO) standards. The development and improvement of this program can benefit from research data published in oral health knowledge of all stakeholders, including teachers, parents, and students, who work together to promote health alongside education.

### Limitations of the Study

The study was limited to teachers in Riyadh, and the findings may not reflect the periodontal and oral health knowledge of teachers in other regions of Saudi Arabia, especially in rural or socio-culturally different areas. Therefore, caution should be exercised when trying to extrapolate these findings to a broader population of teachers in Saudi Arabia. Future research should aim at including stratified sampling across regions to ensure a more diverse sample of teachers and consider guided response collection to enhance generalizability.The questionnaire was distributed online to teachers from the designated schools; reliance on online distribution may limit the number of respondents or exclude teachers who are less experienced in technology.The possibility of self-report bias exists in non-guided questionnaire responses.

## 5. Conclusions

The studied sample of the Government schoolteachers in Riyadh possesses a good level of basic oral health knowledge but has inadequate knowledge regarding periodontal health and disease. Most teachers did not respond correctly to questions on the etiologic factors, prevention, manifestations, consequences, and treatment of periodontal diseases. Factors that significantly influenced teachers’ knowledge were age, gender, region of work, and teaching stage. This information can be utilized by people in authority and decision-making to develop teacher training programs and plan for the allocation of resources. Teachers are very effective in delivering health-related knowledge to school children. Therefore, integrating oral health promotion into school curricula and introducing oral health training for teachers are essential steps to enhance their ability to support and improve students’ overall health.

## Figures and Tables

**Figure 1 ijerph-22-01142-f001:**
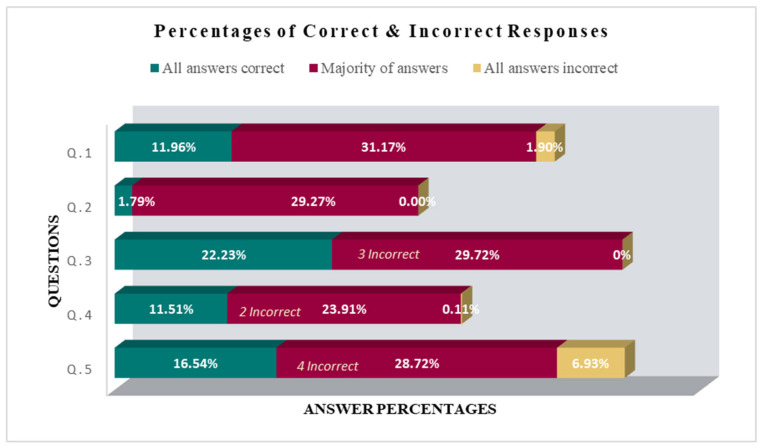
Percentages of Correct and Incorrect Answers.

**Table 1 ijerph-22-01142-t001:** Demographic characteristics of participants.

Demographics	Total	*n* (%)
Gender	Female	465 (52)
	Male	430 (48)
Age	Mean ± SD	40.41 ± 7.314
	Median (Range)	40 (21–64)
	Range	21–64
Marital status	Single	97 (10.84)
	Married	706 (78.88)
	Divorced	69 (7.71)
	Widowed	23 (2.57)
Education	University	780 (87.15)
	Postgraduate Education	115 (12.85)
Stage of teaching	Pre-KG *	32 (3.58)
	KG *	20 (2.23)
	Primary	288 (32.18)
	Intermediate	252 (28.16)
	Secondary	303 (33.85)
Family income (SAR)	<5000	40 (4.47)
	5000/10,000	290 (32.4)
	>10,000	565 (63.13)
Region of work in Riyadh	North	210 (23.46)
	East	235 (26.26)
	South	232 (25.92)
	West	142 (15.87)
	Central	76 (8.49)
Teaching subjects	Arabic language	168 (18.77)
	English language	105 (11.73)
	Sciences	196 (21.9)
	Sports	28 (3.13)
	Music	4 (0.45)
	Social studies	86 (9.61)
	Religion	149 (16.65)
	Computer	61 (6.82)
	Art	72 (8.04)
	Life skills	26 (2.91)

* KG: Kindergarten.

**Table 2 ijerph-22-01142-t002:** Knowledge and awareness of oral and periodontal health and hygiene.

Question	Answers	Participants Responses *n* (%)
To keep my teeth healthy, I should: (select all that apply)	a. Brush my teeth 2–3 times a day *	704 (78.66)
	b. Floss every day *	414 (46.26)
	c. Rinse with a mouthwash *	332 (37.09)
	d. Limit sugary foods *	401 (44.8)
	e. Visit the dentist for regular checkups *	428 (47.82)
2. To keep my gums healthy, I should: (select all that apply)	a. Brush my teeth 2–3 times a day *	518 (57.88)
	b. Have adequate nutrition *	554 (61.9)
	c. Use dental floss *	339 (37.88)
	d. Rinse with salty water	398 (44.47)
	e. Rinse with a mouthwash *	423 (47.26)
	f. Rinse with Myrrh	190 (21.23)
	g. Use special toothpaste for gums	465 (51.96)
	h. Visit the dentist for regular checkups *	374 (41.79)
3. Which of the following is a sign of periodontitis? (select all that apply)	a. Pain *	443 (49.5)
	b. Gum swelling *	575 (64.25)
	c. Gum bleeding *	666 (74.41)
	d. Abscess formation *	474 (52.96)
	e. Tooth mobility *	305 (34.08)
4. Which of the following is a causative factor of periodontitis? (select all that apply)	a. Bacteria on teeth (plaque/biofilm) *	562 (62.79)
	b. Viruses	224 (25.03)
	c. Fungi	412 (46.03)
	d. Tobacco Smoking in any form	437 (48.83)
	e. Diabetes Mellitus	349 (38.99)
	f. Genetic predisposition	252 (28.16)
	g. Pregnancy	129 (14.41)
	h. I do not know	146 (16.31)
5. Periodontal diseases can lead to: (select all that apply)	a. Dental caries	384 (42.91)
	b. Formation of periodontal pockets *	407 (45.47)
	c. Gum recession *	520 (58.10)
	d. Resorption of bone surrounding teeth *	407 (45.47)
	e. Tooth mobility *	418 (46.70)
	f. Tooth loss *	440 (49.16)
6. A periodontal pocket is:	a. An inflammatory process leading to the formation of a space between gums and teeth and bone loss *	323 (36.09)
	b. Gum recession	190 (21.23)
	c. I do not know	382 (42.68)
7. Periodontitis may pose a risk to the general health of a person	a. True *	663 (74.07)
	b. False	55 (6.15)
	c. I do not know	177 (19.78)
8. Gum disease can be treated by: (select all that apply)	a. Cleaning the teeth by a dentist *	453 (50.61)
	b. Taking antibiotics	372 (41.56)
	c. Using a special toothpaste for inflamed gums	591 (66.03)
	d. Using a special mouthwash	364 (40.67)
	e. I do not know	36 (4.02)
9. How frequently should you visit the dentist?	a. When I have pain or any symptoms in the mouth	320 (35.75)
	b. Every 3–6 months or when I have any symptoms in the mouth *	301 (33.63)
	c. Every 6–12 months or when I have any symptoms in the mouth	274 (30.62)
10. I have received dental health education about toothbrushing and flossing from a dentist.	a. Never	202 (22.57)
	b. Once	286 (31.96)
	c. More than once	407 (45.47%)
11. Do you provide your students with any oral health-related information?	a. Yes	360 (40.22)
	b. No	535 (59.78)
12. What learning resources do you use to obtain knowledge about oral health? (select all that apply)	a. My dentist	489 (54.64)
	b. My friends	177 (19.78)
	c. My colleagues at work	166 (18.55)
	d. My family	157 (17.54)
	e. Social media	516 (57.65)
	f. Other	44 (4.92)

*: Correct answer.

**Table 3 ijerph-22-01142-t003:** Distribution of correct answer scores among sociodemographic factors.

Characteristics	Mean Score of Correct Answers	SD	*p*-Value *
Gender			
Male	58.76	11.67	<0.001
Female	61.55	9.49	
Family income (SAR)			
<5000	55.94	10.12	<0.001
5000–10,000	57.68	9.97	
>10,000	61.81	10.75	
Region in Riyadh			
East	58.04	9.01	<0.001
West	59.03	9.74	
North	63.48	11.19	
South	61.19	12.00	
Central	57.12	8.64	
Education			
University	60.23	10.69	0.876
Postgraduate Education	60.07	10.66	
Marital status			
Single	58.92	9.47	0.08
Married	60.57	10.79	
Divorced	57.57	10.28	
Widowed	62.5	11.87	
Teaching Stage			
Pre KG	55.39	9.98	<0.001
KG	54.00	7.27	
Primary	59.59	9.99	
Intermediate	61.35	11.87	
Secondary	60.77	10.25	
Teaching Subjects			
Arabic	60.25	9.34	<0.001
English	62.55	11.09	
Sciences	62.54	10.78	
Sports	55.53	11.81	
Music	61.25	9.46	
Social studies	56.02	8.74	
Religious studies	59.98	10.45	
Computer	59.75	12.21	
Art	60.21	11.21	
Life skills	54.04	10.24	
Overall Knowledge Score	60.21	10.68	

* *p*-value was based on the *t*-test or one-way analysis of variance test to evaluate the independence of sample characteristics and knowledge on oral hygiene.

**Table 4 ijerph-22-01142-t004:** Results of multiple regression analysis with the scores as the outcome variable.

	B (SE of B)	β (*t*-Value)	*p*-Value
Age	0.278 (0.054)	0.190 (5.179)	<0.001
Marital status			
Single	Ref		
Married	−2.821 (1.233)	−0.107 (−2.29)	0.0224 *
Divorced	−4.708 (1.671)	−0.118 (−2.818)	0.0049 *
Gender			
Female	Ref		
Male	−2.214 (0.736)	−0.103 (−3.01)	0.0027 *
Region of work in Riyadh			
North	Ref		
East	−3.965 (0.999)	−0.163 (−3.967)	0.0001 *
West	−2.792 (1.131)	−0.095 (−2.468)	0.0138 *
Central	−4.446 (1.391)	−0.115 (−3.196)	0.0014 *
Teaching stage			
Pre school	Ref		
Intermediate	4.062 (1.979)	0.171 (2.052)	0.040 *

*: Statistically significant; B: unstandardized regression coefficient; SE of B: standard error; β (*t*-value): standardized coefficient; *t*-value: *t*-statistic.

## Data Availability

The data presented in this study are available on request from the corresponding author due to Institutional data confidentiality restrictions.

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
