# Peer review of "Periodontal Health Knowledge of Schoolteachers: A Cross-Sectional Study"

_ijerph, 2025, doi:10.3390/ijerph22071142_

Round 1

Reviewer 1 Report

Comments and Suggestions for Authors

This paper has significant purpose relative to the issues of periodontal knowledge and prevention.  Perhaps questions about "diet and nutrition" could have been included in the questionnaire.  Explanation of the teacher's sample size was impart and explained adequately.  Being a parent of one's own children would have been impactful within the questionnaire, especially after that question was raised in the discussion.  

Line 154 the word "pause" should be replaced.  I breleive it should be "pose"

Live 196 check grammar

Within the questionairre, was there a section per question as "I do not know"?

Good information obtained.  How will it be the information be used?  Will it enhance prevention training for teachers, students and parents?

Reviewer 2 Report

Comments and Suggestions for Authors

This is an interesting study, but there are some points of concern, which are noted below.

Title: Periodontal health is a term included in Oral health. Is it necessary to list the terms Oral health and Periodontal health together?

Informed consent statement: It is stated that the authors obtained consent from all subjects (lines 341-342). How did you actually obtain consent?

Questionnaire:

  1. It is stated that the authors used a validity and feasibility questionnaire (lines 91-93); how was validity and feasibility verified? If the authors used their original questionnaire, please describe it in more detail.

  1. Am I correct in understanding that you mailed the survey form and accepted responses via Google form?

  1. Are there any criteria for selecting the questions? Seven of the 12 items were related to gingival health, which appears to be a skewed theme for a questionnaire that asks about knowledge of oral and periodontal health. The authors could have asked for knowledge about caries and mastication as part of oral health knowledge.

  1. In question 9, a dental visit every 3-6 months is more correct than a dental visit every 6-12 months, but if the subject is in good oral health, then a dental visit every 6-12 months would be fine.

Statistical analysis:

  1. What criteria were used for the confounding factors listed in Table 1? In addition, how were these data obtained?

  1. A one-way analysis of variance was used, but were the positive answer scores normally distributed?

Table 2: question 9a  Please change the words from “rue” to “true”.

Table 4: What is the goodness of fit of the multiple regression analysis model?

Reviewer 3 Report

Comments and Suggestions for Authors
  1. Regarding Table-1:
    1. Why median age was calculated, why not mean age? This indicates perhaps the data was not normal.
    2. The difference between categories "single", "divorced" and "widowed" is not clear. Are not "divorced" and "widowed" also considered "single"?
    3. The difference between "university" and "higher university" is not clear; are not these the same?
    4. On what basis the categorization done for education: primary, intermediate, and secondary?
    5. On what basis family income was broken down into three categories?
  2. Referring to table-2
    1. "definition" of "periodontal pocket" is a very precise, and obviously with a leading structure. This may be the reason for increased response marked by the respondents. What other justification do the authors have in defense that the questions and options provided were well-structured?
    2. check typo error in Q-7.
  3. What was the normality assumption to run the test of linear regression? 
  4. What was the level of significance considered to keep the variable in linear regression model? Why "marital status" with p-value = 0.08 and not "family income" with p-value <0.001 was considered to be included in the multiple regression model?
  5. Which type (hierarchical or stepwise - forward / backward) of multiple regression was used?
  6. The conclusion for "Government school teachers" cannot be justified unless studied in reference to Non-government school teachers. Either rephrase your conclusion or justify the existing conclusion.

Reviewer 4 Report

Comments and Suggestions for Authors

Dear Authors,

We commend you for addressing an important and relevant topic in public health. Your study on the oral and periodontal health knowledge of schoolteachers in Riyadh is timely and insightful.

After careful review, we believe the manuscript requires major revisions before it can be considered for publication. Specifically, we recommend the following:

Improve the English language and grammar throughout the manuscript, preferably with assistance from a native speaker or professional editor.

Include the full questionnaire instrument as an appendix or supplementary file.

Provide more details regarding the validation of the questionnaire, especially construct validity and reliability testing.

Clarify whether logistic regression was used, and present the results if applicable.

Revise the conclusions to more accurately reflect the findings, particularly regarding periodontal knowledge.

Include in Discussion chapter comparisons with other regional or international studies.

Elaborate on the ethical protocol regarding informed consent and data confidentiality.

We believe the manuscript has strong potential once these revisions are addressed and look forward to reviewing a revised version.

Reviewer 5 Report

Comments and Suggestions for Authors

The topic is relevant and of potential interest to the field. However, the manuscript presents several conceptual, methodological, and reporting issues that need to be addressed to meet the standards of academic publication. The article requires a thorough revision before it can be published, as it contains significant gaps in the methodology and presentation of the results.

Round 2

Reviewer 2 Report

Comments and Suggestions for Authors

The authors have addressed my sugessitions adequately.

Author Response

We sincerely thank the reviewer for their thoughtful comments and valuable suggestions, which have greatly helped improve the quality and clarity of our manuscript. 

Reviewer 4 Report

Comments and Suggestions for Authors

Dear authors,

I see you have fulfilled my requests, and now the article is improved and ready for publication. I wish you success in your professional life and research field!

Author Response

We sincerely thank the reviewer for their efforts, thoughtful comments, and valuable suggestions, which have greatly helped improve the quality and clarity of our manuscript. We are also very grateful for the kind wishes.

Reviewer 5 Report

Comments and Suggestions for Authors

The authors have taken into consideration several of the previous suggestions and comments, which is appreciated. However, some important issues still require clarification and adjustment:

  • The manuscript should explicitly state in the methodology how the knowledge score was calculated. It is also necessary to indicate what the scores represent in terms of knowledge levels.

  • The manuscript refers to non-parametric tests in the methodology, but it does not appear that these tests were actually used. If non-parametric tests were not employed, this reference should be removed to avoid confusion.

  • The limitations should be presented at the end of the discussion section rather than as a separate chapter. Furthermore, this section should emphasise how the identified limitations may affect the generalizability or extrapolation of the study's findings.

  • When reporting differences between groups in the results section, it is essential to specify which groups differ from one another. Currently, if there are more than two groups, it is unclear which specific comparisons yielded significant differences.

Author Response

We are grateful to the reviewer for their efforts, as well as detailed and insightful feedback.  We have carefully addressed each comment and revised the manuscript accordingly.  
